# Identification of Diverse Stress-Responsive Xylem Sap Peptides in Soybean

**DOI:** 10.3390/ijms23158641

**Published:** 2022-08-03

**Authors:** Wai-Ching Sin, Hon-Ming Lam, Sai-Ming Ngai

**Affiliations:** Center for Soybean Research of the State Key Laboratory of Agrobiotechnology and School of Life Sciences, The Chinese University of Hong Kong, Hong Kong, China; sin-9527@hotmail.com (W.-C.S.); honming@cuhk.edu.hk (H.-M.L.)

**Keywords:** xylem sap, peptide signaling, long-distance, root-to-shoot, peptidomic, C-terminally encoded peptide, sulfated peptide, CLV3/ESR-related, post-translational modification

## Abstract

Increasing evidence has revealed that plant secretory peptides are involved in the long-distance signaling pathways that help to regulate plant development and signal stress responses. In this study, we purified small peptides from soybean (*Glycine max*) xylem sap via *o-c*hlorophenol extraction and conducted an in-depth peptidomic analysis using a mass spectrometry (MS) and bioinformatics approach. We successfully identified 14 post-translationally modified peptide groups belonging to the peptide families CEP (C-terminally encoded peptides), CLE (CLAVATA3/embryo surrounding region-related), PSY (plant peptides containing tyrosine sulfation), and XAP (xylem sap-associated peptides). Quantitative PCR (qPCR) analysis showed unique tissue expression patterns among the peptide-encoding genes. Further qPCR analysis of some of the peptide-encoding genes showed differential stress-response profiles toward various abiotic stress factors. Targeted MS-based quantification of the nitrogen deficiency-responsive peptides, GmXAP6a and GmCEP-XSP1, demonstrated upregulation of peptide translocation in xylem sap under nitrogen-deficiency stress. Quantitative proteomic analysis of *GmCEP-XSP1* overexpression in hairy soybean roots revealed that *GmCEP-XSP1* significantly impacts stress response-related proteins. This study provides new insights that root-to-shoot peptide signaling plays important roles in regulating plant stress-response mechanisms.

## 1. Introduction

Terrestrial plants require constant adaptation to the changing environment. To maintain quick and systematic responses to environmental changes, plants utilize a variety of signaling molecules that are involved in specific signaling pathways to coordinate the optimum shoot–root balance and development strategies. The vascular tissues of plants play a crucial role as long-distance-traveling conduits for long-distance signaling throughout the plant body [1,2]. Decades of research have shown that a wide range of biological molecules are employed in long-distance signaling in plants, including phytohormones, secretory peptides, calcium, and phloem mobile mRNA [2,3,4]. Among the signaling strategies, plant signaling peptides work as the primary messengers involved in the regulation of plant development, the coordination of stress responses, and pathogen interactions. The signaling peptides in plants are typically post-translationally modified oligopeptides that carry characteristic post-translational modifications (PTMs), such as tyrosine sulfation, proline hydroxylation, and proline triarabinosylation [5,6]. There is increasing awareness that plant secretory peptides can be translocated through the xylem stream to mediate root-to-shoot communication under biotic and abiotic stresses [4,7].

Previous research has shown that the root-to-shoot-translocated peptides in plants are broadly involved in the adaptations and development of plants in response to systemic stress. In *Lotus japonicus*, on rhizobia infection, the CLE-root signal (CLE-RS) genes, *LtCLE-RS1* and *LtCLE-RS2*, are upregulated in the roots to initiate a long-distance negative feedback loop for the autoregulation of nodulation [8]. The mature form of the CLE-RS2 peptide has been demonstrated to travel as a root-ascending messenger via xylem sap and physically interact with the shoot receptor HAR1, which promotes the expression of the shoot cytokinin synthesis gene *IPT3*, eventually increasing cytokinin shoot-to-root transportation to inhibit root nodulation formation [8,9]. In *Arabidopsis thaliana*, *CEP* genes in the roots were found to be upregulated under nitrogen deficiency stress, and the secreted peptides travel through the xylem stream to interact with shoot CEP Receptor 1 (CEPR1). The CEPs–CEPR interaction subsequently triggers the shoot-derived signal CEP Downstream 1 (CEPD1) and CEPD2 translocation from shoot to root to modulate root nitrogen transporter gene expression [10,11]. Similarly, in *Arabidopsis thaliana,* when experiencing drought stress, the root-derived CLE25 peptide is transported from root to shoot to regulate stomatal closure in the leaves in association with Barely Any Meristem (BAM) receptors [12]. Nonetheless, despite consensus regarding the existence of an organ-to-organ communication network in plants, long-distance signaling in plants remains an underexplored field. There are many long-distance signaling peptides that have yet to be discovered, and the mechanisms of action of these long-distance secretory peptides remain elusive.

Plant xylem sap is an apoplastic fluid with functional roles and that carries water, nutrients, and metabolites from root to shoot through xylem vessels. Xylem sap comprises a vast diversity of organic compounds, and many studies have detected a complex xylem sap profile that includes a number of organic substances, including amino acids, organic acids, proteins, carbohydrates, lipids, and phytohormones [13,14,15,16]. Whether the contents of xylem sap provide any biological functions in plants has long been questioned. From the perspective of the plant environmental response, a quantitative proteome analysis of xylem sap under abiotic and biotic stresses, such as nitrogen status and pathogenic interaction, has shown dynamic changes in the protein profile that may allow stress adaptations and antimicrobial functions [17,18,19,20,21]. When looking at the “micro” molecules in xylem sap more closely, xylem sap transports small oligopeptides that serve as essential messengers for the maintenance of systemic homeostasis in plants. Until recently, only a few peptidomic experiments focusing on xylem sap had been performed, for instance, in *Glycine max* and *Medicago truncatula* [22,23].

High-throughput liquid chromatography-tandem mass spectrometry (LC-MS/MS) is a current and effective method to analyze post-translationally modified peptides with high sensitivity and specificity. However, most plant secretory peptides have a low native concentration, which makes LC-MS-based peptidomic analysis from biological material highly difficult. Such difficulty can be caused by (1) a low peptide abundance that falls below the detection threshold of the instrument; (2) the incomplete separation of peptides by liquid chromatography; (3) a strong background from small molecules and degraded protein products that masks the secretory peptide signals. Partial chemical purification using the *o-c*hlorophenol/acetone method followed by advanced LC-MS analysis, which was introduced by Ohyama et al., represents an effective method to enrich small peptides in solution [24]. This novel chemical method has enabled the enrichment of small peptides in plant fluid through the partial removal of the background created by small molecules, thereby largely improving downstream MS peptide analysis. In a study of glycopeptide LtCLE-RS, *o-c*hlorophenol extraction was used to recover the root-derived LtCLE-RS peptides in the xylem sap of transgenic soybeans [8]. The application of the *o-c*hlorophenol/acetone extraction method combined with a targeted MS approach was also successfully implemented to identify and quantify *MtCEP1* peptide variants from a hairy root culture of *Medicago* overexpressing *MtCEP1* [25]. Moreover, in the peptidomic analysis of xylem sap from *Glycine max* and *Medicago truncatula*, similar approaches were also adopted that enriched the signaling peptides in plants [22,23].

In the present investigation, we harvested soybean xylem sap and enriched small peptide fractions using the *o-c*hlorophenol extraction method. We conducted an LC-MS-based peptidomic analysis of the small peptides in soybean xylem sap and successfully detected 14 post-translationally modified peptide groups corresponding to the CEP, CLE, and PSY peptide families. We examined the tissue expression pattern of the peptide precursor genes by qPCR to locate root-derived peptides. From the root-derived peptide genes, we assessed the stress responsiveness of the various peptide precursor genes via qPCR, and we found peptide genes that were sensitive to the nitrogen status, drought, and iron deficiency. We then focused on two nitrate-sensitive CEP peptides, GmCEP-XSP1 and GmXAP6a, for further analysis. We conducted relative peptide quantification using targeted MS and confirmed an upregulated root-to-shoot translocation of the two CEP peptides in xylem sap under low-nitrogen treatment conditions. The biological activity assay carried out using synthetic CEP peptides showed an inhibitory effect on primary root growth in both the GmCEP-XSP1 and GmXAP6 treatment groups. We generated *Agrobacterium rhizogenes*-mediated hairy root soybean transformants overexpressing GmCEP-XSP1 for quantitative proteomic analysis. GO enrichment analysis revealed robust enrichment of the stress response-related GO terms in the biological process, showing that the expression of the xylem sap peptide can confer a systemic stress response. The results of our study may help to broaden our horizons in the investigation of the biological functions and mechanisms of long-distance signaling in plants.

## 2. Results

### 2.1. Purification and Identification of Soybean Xylem Sap Oligopeptides

Soybean plants were cultured in hydroponic conditions until the V3 growth stage, in which the third trifoliate was fully unfolded. We then collected xylem sap fluid from the cut surface of the soybean hypocotyls via the root pressure method. Initial SDS-PAGE analysis indicated that the collected soybean xylem sap samples contained many high-molecular-weight proteins (Appendix A). To identify the small oligopeptides in the soybean xylem sap, the collected soybean xylem sap samples were first purified using the *o-c*hlorophenol/acetone extraction method [24]. Through removal of the high-molecular-weight proteins, small peptide fractions (molecular weight <10,000) were enriched using Amicon Ultra 10K MWCO filters (Merck Millipore, Carrigtwohill, Ireland) as a filtrate. The small peptide fractions were further purified and fractionated using a high-pH fractionation kit. A data-dependent MS/MS analysis was performed with an Orbitrap mass spectrometer to analyze the soybean xylem sap peptidome. The collected MS/MS spectra were searched for in soybean databases using Proteome Discoverer (version 2.4) for peptide matching and PTM analysis.

Peptide matching yielded a total of 6299 peptide groups matching 1334 proteins. Despite the initial enrichment of the small peptide fraction, the peptide pool still mostly consisted of the degraded peptide fragments of larger proteins. The amino acid length of the identified proteins ranged from 53 to 5108 aa, with over 90% of the proteins below 1000 aa long (Appendix A). The precursors for small secretory peptides in plants are generally reported as polypeptides that are <150 aa in size. After cleavage from the precursor polypeptides, the biologically active peptide domain is post-translationally modified via proline hydroxylation, proline triarabinosylation, and tyrosine sulfation [5]. Based on this information, the search criteria were adjusted to target proteins that were <200 aa in size to narrow down potential signaling peptide candidates. This led us to identify 14 peptide groups matching a total of 22 precursor genes from the soybean xylem sap peptidome (Table 1). These included six of the soybean XAP peptide groups (GmXAP1–6) that were first reported by Okamoto et al. [22]. Additionally, we found a total of eight novel peptide groups that fulfilled our criteria, and we named the newly discovered peptides xylem sap peptides (XSP).

### 2.2. Characteristics of Peptides Identified in Xylem Sap

Based on the structural features of the sequences, we categorized the peptides into groups according to whether they were CEP, PSY (peptide with sulfated tyrosine), or CLE peptides (Table 1, Appendix A). Combined with their corresponding peptide class, the newly identified peptides were named CEP-XSP 1-3, PSY-XSP 1-4, and CLE-XSP1. We then named the peptides that corresponded to multiple precursor polypeptides PSY-XSP3a and PSY-XSP3b, CLE-XSP1a and CLE-XSP1b, etc. (Table 1).

A total of four peptide groups (GmXAP6 and GmCEP-XSP1–3) were included in the CEP category. The peptides ranged from 14 to 19 aa in length and were grouped by sequence features resembling those of the CEP family (Appendix A). The MS spectra-matching results showed that all of the CEP peptides that were identified contain hydroxylated proline residues in the range of one to three modified residues (Table 1). The precursor polypeptide sequences of GmXAP6a-c and GmCEP-XSP1 containd two peptide domains, which we refer to as domain 1 (D1) and domain 2 (D2) (Table 1, Appendix A). Overlapping peptide domain sequences with little difference in the residues were observed among the two domains containing CEPs (Appendix A). For GmXAP6a and GmCEP-XSP1, we detected identical D1 peptide sequences with length variants of 14 and 16 aa (Table 1). For GmXAP6a–c, the D2 sequences overlap for the first 15 residues but differ at the 16th residue. We could detect the 16 aa-long D2 peptide sequences of GmXAP6a, GmXAP6b, and GmCEP-XSP1, but not GmXAP6c (Table 1). GmCEP-XSP1 contains a unique 16aa D2 sequence that was detected via the PTM of Hyp11 and Hyp7,11. GmCEP-XSP2 and Gm-CEP-XSP3 are the CEP members with only one peptide domain. GmCEP-XSP2 was identified as a 14 aa-long peptide with three proline hydroxylation sites, and the peptide sequence was matched to two precursor genes. For GmCEP-XSP3, peptide-length variants of 15 and 19 aa were detected with proline hydroxylation.

A total of eight peptide groups (GmXAP1–3, GmXAP5, and GmPSY-XSP1–4) are included in the PSY category, with peptide lengths varying from 16 to 21 aa. This group was categorized by the DY motif at the N-termini, which is a characteristic sulfation site of sulfated peptides [26]. All of the identified oligopeptides in this group were identified with a sulphated tyrosine residue at position 2, but peptides with an unmodified Y2 were also detected (Table 1). Sequence alignment with sulfated peptides from *Arabidopsis* suggested that the identified peptides are homologs to the PSY family, sharing the structural characteristic of the N-terminal DY motif and amino acid asparagine at position 8 (Appendix A). Meanwhile, GmXAP3 was found to be a homolog to *Arabidopsis* Casparian Strip Integrity Factor (CIF) peptides, a subfamily of sulfated peptides (Appendix A) [27]. Length variants were observed in the individual peptides GmXAP3, GmXAP5, GmPSY-XSP1, and GmPSY-XSP3. Peptide-matching results showed that GmPSY-XSP1 is also a glycopeptide that is modified by triarabinosylation on proline residues.

Two peptides with a length of 12 aa and that matched three precursor genes were identified to be CLE members. In this group, GmXAP4 and GmCLE-XSP1 are CLE peptides that differ by only 1 aa at position 2 (Appendix A). Both peptides were identified with PTM patterns of proline hydroxylation and proline triarabinosylation. In our spectrum-matching results, the GmXAP4 peptide demonstrated proline triarabinosylation at position 7, while in GmCLE-XSP1, proline triarabinosylation was detected at position 4.

### 2.3. Tissue Expression Pattern Analysis

To discover the potential root-to-shoot signaling candidates, we next examined the tissue expression patterns of polypeptide genes via RT-qPCR. From soybean plants in the V1 growth stage, the stage in which the first trifoliate is fully unfolded, we collected the leaves, second internode (between the first trifoliate and unifoliate), hypocotyl, and root tissues for RNA extraction and cDNA synthesis. We compared the relative expression levels between tissue types. Diverse tissue expression patterns were observed among the peptide precursor genes in the CEP and PSY groups, but not in the CLE group, which was predominantly expressed in the root tissue (Figure 1). To identify root-to-shoot communication molecules, we focused on the genes that are highly expressed in root tissues.

### 2.4. Abiotic Stress Response Pattern of Xylem Sap Peptide Genes

Long-distance signaling plays a crucial role in the systemic stress adaptations of plants by transmitting stress signals from sensing tissues to distant organs [2,7]. To understand the involvement of peptide-encoding genes in root-to-shoot signaling, precursor genes that are highly expressed in root tissues were selected for stress response analysis. Next, we placed the soybean seedlings under various abiotic stresses, including nitrogen deficiency, nitrogen oversupply, salinity, PEG-induced drought, phosphate deficiency, and iron deficiency. We then collected the root tissue samples for RT-qPCR analysis to examine the gene expression levels (Figure 2). *GmCEP-XSP1*, *GmXAP6a,* and *GmXAP6b* were found to be significantly upregulated during nitrogen deficiency. In contrast, *GmXAP1*, *GmXAP2*, *GmXAP4*, *GmCLE-XSP1a,* and *GmCLE-XSP1b* were significantly downregulated during nitrogen deficiency. The expression of *GmXAP3a*, *GmXAP3b*, *GmPSY-XSP1*, and *GmPSY-XSP3* was significantly upregulated during PEG-induced drought stress. *GmXAP3a*, *GmCLE-XSP1a,* and *GmCLE-XSP1b* were significantly upregulated under iron deficiency. Meanwhile, *GmXAP1*, *GmCEP-XSP1*, *GmCEP-XSP2, GmXAP6a,* and *GmXAP6b* were significantly downregulated under iron deficiency. Our RT-qPCR results indicated diverse stress response patterns among the xylem sap peptide genes at the transcript level, suggesting the broad involvement of xylem sap peptide genes in abiotic stress responses.

### 2.5. GmCEP-XSP1 and GmXAP6a Root-to-Shoot Translocation under Nitrogen Deficiency

Based on our RT-qPCR results, the expression levels of *GmCEP-XSP1* and *GmXAP6a* were increased during the nitrogen deficiency test (Figure 2B). To verify the root-to-shoot translocation of the GmCEP-XSP1 and GmXAP6a peptides, targeted MS quantification was performed. Parallel reaction monitoring (PRM) is a targeted MS technique that is applied in new-generation quadrupole mass spectrometers to achieve sensitive and accurate peptide quantification [28,29]. We collected xylem sap from soybeans that had been exposed to control and nitrogen-starved treatments for 48 h. A relative quantification of the GmCEP-XSP1 and GmXAP6a peptides in xylem sap under the control and nitrogen-starved conditions was performed by PRM analysis. The extracted peptide ion signals of GmCEP-XSP1 and GmXAP6a were quantified using Skyline software [30]. We scanned the possible proline hydroxylation states of the GmCEP-XSP1 and GmXAP6a peptides (Appendix A). For GmCEP-XSP1/GmXAP6a-D1, we targeted the 14 aa-long peptide domain as it was detected in a higher abundance of spectra in the initial peptidomic analysis. We could successfully detect GmCEP-XSP1/GmXAP6a-D1 Hyp4 and Hyp7, Hyp11, along with Hyp7, Hyp11 modification variants, but we could not consistently obtain stable MS peak-area signals within the technical replicates (data not shown). For GmCEP-XSP1-D2 and GmXAP6a-D2, we only detected GmCEP-XSP1-D2:Hyp11 and GmCEP-XSP1-D2:Hyp7; Hyp11 modification variants; GmXAP6a-D2:Hyp11 (Figure 3). Relative quantification by the peak area of the MS2 spectra showed that the GmCEP-XSP1 and GmXAP6a peptides had increased abundance in the xylem sap under low-nitrogen conditions (Figure 4, Appendix A). Our results showed evidence in support of the notion that the nitrogen status-sensitive GmCEP-XSP1 and GmXAP6a peptides are root-to-shoot translocated in xylem sap.

### 2.6. Synthetic Peptides of GmCEP-XSP1 and GmXAP6 Inhibit Primary Root Growth

Based on the modification variants confirmed from the MS spectra in the PRM analysis, we tested the biological activity of the modified synthetic peptides GmCEP-XSP1/GmXAP6a-D1 (Hyp 7,11 and Hyp 4,7,11), CEP-XSP1-D2 (Hyp11 and Hyp 7,11), and GmXAP6-D2 (Hyp11) (Figure 5A). We treated 4-day-old soybean plants with a culture solution supplemented with synthetic peptides for 7 days. In our results, treatment with the D1 peptides of GmCEP-XSP1 and GmXAP6a did not yield significant results for soybean primary root growth, except for GmCEP-XSP1/GmXAP6a-D1 Hyp11 (Figure 5B). The modified D2 peptides from CEP-XSP1 and XAP6 showed significant primary root growth inhibition in the soybean seedlings. For CEP-XSP1-D2, Hyp 7,11 modifications showed a more potent inhibitory effect than Hyp11 modifications (Figure 5C).

### 2.7. Proteome Analysis of Soybean Transgenic Hairy Root GmCEP-XSP1ox

Our results suggest that GmCEP-XSP1 and GmXAP6a are nitrogen-deficiency-responsive peptides that are mobile in xylem sap and can alter soybean root growth (Figure 2, Figure 4, and Figure 5). To gain a overall understanding of the potential functional roles of the nitrogen-responsive CEP peptides in soybean, we generated an *Agrobacterium*-mediated soybean hairy root transformant overexpressing GmCEP-XSP1 for a gain-of-function analysis. We quantified the differences in the root proteomes in control and GmCEP-XSP1ox soybean hairy root tissue using label-free MS-based quantification (Figure 6). A total of 7285 protein groups were identified in the protein-matching results. A total of 428 proteins were found to have significant differences (adjusted *p*-value < 0.05), including 303 upregulated and 125 downregulated proteins. To gain an overall understanding of the proteomic changes, we analyzed the gene ontology (GO) annotations for the differentially expressed protein (DEP) genes via GO term enrichment analysis. The results indicated that the top-four (ranked by -log FDR) enriched biological process GO terms were “response to stimulus”, “response to stress”, “secondary metabolic process”, and “response to external stimulus” (Figure 7). The construction of hierarchy trees depicting the GO terms showed that the four major clusters of GO terms were “response to stimulus”, “organic substance metabolic process”, “nitrogen compound metabolic process”, and “cellular metabolic process” (Appendix A). Our results indicated that CEP-XSP1 overexpression induced a significant change in the root proteome related to the stress response and the metabolic processes of the macromolecules.

To further investigate the potential relationship between GmCEP-XSP1 and the response to nitrogen-deficiency stress in soybean, we generated cDNA from control and GmCEP-XSP1ox soybean hairy root tissue for a qPCR analysis of the soybean nitrate-transporter genes *GmNRT1.1*, *GmNRT1.2*, *GmNRT1.3*, *GmNRT1.5*, and *GmNRT2.0*. In our results, we observed a significant upregulation of the nitrate transporters *GmNRT1.1*, *GmNRT1.2*, and *GmNRT1.3* and downregulation of *GmNRT1.5* and *GmNRT2.0* (Figure 8). This suggests that GmCEP-XSP1 may influence the nitrogen-uptake process in soybean in association with nitrate transporters. Taken together, we propose that GmCEP-XSP1 may play an important role in abiotic stress-signaling in soybean under nitrogen-deficient conditions.

## 3. Discussion

Large plant models such as broccoli (*Brassica oleracea*), soybean (*Glycine max*), rice (Oryza sativa), and legumes (*Medicago truncatula*) were frequently studied in early xylem sap proteomic research, but many of the frontier studies relied on gel-based approaches that limited the analysis of small and low-abundance peptides [15,31,32,33,34]. With the advancement of LC-MS/MS systems, the identification of post-translationally modified small peptides in plant xylem sap has become achievable, expanding our understanding of long-distance signaling. In this study, we utilized an *o-c*hlorophenol purification system combined with nano LC-MS methods to study the peptidomic profile of soybean xylem sap, which led to the discovery of novel long-distance signaling peptides. From the peptidomic results, we identified 14 peptide groups with signaling peptide characteristics, including eight novel peptide groups that have not been reported in previous soybean xylem sap studies (Table 1). High variation in the PTM patterns was also identified on the peptide sequences, which may suggest some form of peptide regulation.

In our peptidomic profiling of soybean xylem sap, we identified peptides that share sequence similarities with the following plant peptide families: CEPs, CLEs, and PSYs (Appendix A). There is emerging evidence supporting the concept of mobile long-distance signaling in xylem sap from plant peptide families such as CEP and CLE [2]. However, when examining the tissue expression patterns of xylem sap peptides, highly distinct tissue expression patterns were found (Figure 1). While it is tempting to speculate that xylem sap peptides are universally involved in long-distance signaling, signaling peptides in plants are diversely expressed across plant tissues and involved in a wide spectrum of plant development activities [35]. An extensive case-by-case investigation is required to verify the long-distance signaling peptides in plants.

A growing body of evidence indicates that long-distance peptide signaling pathways are broadly involved in the stress-response process in plants [7]. Based on the tissue expression patterns observed in the present study, we focused on peptides that were highly expressed in the root region, which we speculated to be potential root-to-shoot signaling candidates for stress-response analyses (Figure 1). In the results of the qPCR analysis conducted on soybean roots under a variety of stress factors, we detected a diverse spectrum of stress responses in each peptide-encoding gene (Figure 2).

Studies have highlighted the functional roles of CEP peptides as regulators of shoot and root development in plant stress adaptation [36]. Additionally, microarray- and RNAseq-based gene expression analyses of CEPs across several plant species showed that CEP members are diversely regulated in response to different stress conditions [37]. More than growth regulators, plant CEPs may play an active role in different stress-response mechanisms by promoting the pathways necessary for stress adaptation. For instance, *Arabidopsis* CEP peptides were found to be strongly upregulated in the root region in response to nitrogen starvation and to trigger systemic nitrogen uptake regulation via long-distance signaling mechanisms [10,11]. In our study, *GmXAP6a* and *GmCEP-XSP1* expression levels were strongly upregulated by nitrogen deficiency (Figure 2B). Peptide biological activity assays of the synthetic peptides GmXAP6a and CEP-XSP1 suggested that the peptides may negatively regulate root growth in soybean (Figure 5). Out PRM assay also supported an increase in the abundance of GmXAP6a and CEP-XSP1 D2 peptides in xylem sap collected from nitrogen-starved soybean, indicating that they are root-to-shoot translocated peptides (Figure 4). Therefore, we hypothesized that the GmXAP6a and CEP-XSP1 peptides function as the signaling messengers to regulate the nitrogen-deficiency response in soybean. We overexpressed GmCEP-XSP1 via transgenic hairy roots for label-free protein quantification analysis, to help understand the physiological implication of the nitrogen deficiency-induced CEP. GO term analysis revealed a robust proteome change toward stress responses, organic substances, and the metabolic processes of nitrogen compounds (Appendix A). In *Arabidopsis*, the overexpression of *AtCEP5*, which mediates the drought stress response, also had a significant impact on abiotic stress-related proteomic changes, as revealed by quantitative proteomic analysis, pointing to CEPs holding governing roles in various plant stress-response mechanisms [38].

In our results, *GmXAP3a/b*, *GmPSY-XSP1*, and *GmPSY-XSP3* were found to be significantly upregulated under PEG-induced drought stress (Figure 2A). It was demonstrated that root-derived AtCLE25 can regulate stomatal closure in leaves via the BAM receptor, which suggests the possibility of the plant drought-response mechanism using long-distance signaling pathways [12]. Still, to our knowledge, no literature has supported the idea that the sulfated peptides in plants are involved in long-distance signaling, despite their detection in xylem sap. The GmXAP3 peptide is homologous to the sulfated peptide AtCIF1/2 (Appendix A). AtCIF1/2 peptides were identified as root stele-expressed peptides that regulate Casparian strip formation, with long-distance translocation traits reported [27,39]. For the rest of the sulfated peptide candidates, sequence alignment suggests that they are likely to be homologs of PSY family members (Appendix A). Investigation of the *Arabidopsis* PSY1 signaling pathway indicated that the interaction of PSY1 with the PSY1 receptor (PSY1R), a leucine-rich repeat receptor kinase, triggers the downstream activation of the plasma membrane proton pump, AHA2, resulting in proton efflux from the plasma membrane [40,41]. Up to now, our understanding of the PSY family in plants has remained narrow. A study of eight PSY gene homologs (*PSY1-8*) in *Arabidopsis* showed that despite the high conservation in the active peptide domains, there are, in fact, distinct tissue-expression patterns and differential expression toward various stress factors among *PSY* genes [42]. Whether the sulfated peptides found in plants are involved in long-distance signaling in response environmental stimuli, such as drought stress, is interesting to explore. However, for abiotic stresses such as drought and high salinity that involve a drastic change in osmotic pressure, it is not feasible to employ conventional “root pressure” sampling methods to collect sufficient xylem sap samples from stressed plants; instead, the application of external pressure via a custom-built pressure chamber or Scholander-type chamber may be required [43,44,45]. We did not continue our study on the response of the sulfated peptides to drought stress due to equipment limitations.

GmCLE-XSP1 is a 12 aa-long glycosylated CLE peptide that is structurally similar to AtCLE25 and involved in long-distance signaling during the drought response (Appendix A) [12]. We found that *GmCLE-XSP1a/b* showed a significant increase in its expression when exposed to the iron-deficiency treatment (Figure 2C). Recently, a novel peptide family, *Iron Man* (IMA) or *FE-Uptake-Inducing Peptide* (FEP), which regulates plant iron uptake, was identified by two independent research groups, proving that plant peptide signaling is also involved in the response to iron deficiency [46,47]. This peptide family is highly sensitive to iron deficiency, with the IMA/FEP homologs observed to be strongly upregulated in low-iron conditions. IMA/FEP genes appear to positively regulate iron uptake in plants, with overexpression causing increased root iron uptake and the activation of other iron-deficiency response genes, while gene silencing led to a drastic reduction in iron uptake activity [46,47]. Taken together, these suggest the possibility that the GmCLE-XSP1 peptide participates in the iron deficiency response, perhaps by root-to-shoot translocation. LC-MS/MS analysis identified GmCLE-XSP1 with variants of proline hydroxylation or triarabinosylation on the P4 residue of the peptide domain (Table 1). Our group could not access a commercial supplier that could provide triarabinosylated peptides. We attempted to study the biological activity of GmCLE-XSP1 using synthetic GmCLE-XSP1 peptides with hydroxylated proline residues only, but we could not observe any significant differences in the treated plant, possibly due to the lack of triarabinosylation (Appendix A). A study followed an in-house-developed protocol to chemically synthesize the arabinose chain on *Arabidopsis* CLAVATA3(CLV3), a glycopeptide that regulates the stem cell fate of shoot apical meristem (SAM) [48]. The study compared unmodified, mono-, di-, and tri-arabinosylated CLV3 and demonstrated a progressive increase in the biological activity of peptides along with the arabinose chain length. Additionally, in the same study, a structural analysis of arabinosylated CLV3 by nuclear magnetic resonance (NMR) spectroscopy revealed that the arabinose chain causes conformational changes in the peptide backbone. These results strongly indicate the physiological significance of triarabinosylation for plant peptide activity.

Plants continuously conduct long-distance signaling via their vascular structure to regulate the whole-plant physiological state, to adapt to the changing environment and balance organ development. With technological improvement and cumulative effort by the research community, various xylem sap signaling peptides have been discovered, revealing the vast diversity of the signaling peptides involved in the stress response [4,8,10,12]. In this study, we enriched and purified small peptides in soybean xylem sap and identified novel mobile xylem peptide candidates belonging the CEP, CLE, and PSY families. Our GmCEP-XSP1 results support the physiological implications of the mobile peptides in xylem sap in conferring stress-adaptation processes. A more in-depth investigation is required in the near future to decipher their mode of action. For the rest of the potential root-to-shoot peptide candidates, distinct stress-response patterns were observed, suggesting the possibility of previously unknown signaling pathways. Whether they mediate long-distance signaling and what their biological functions might should be further explored to understand their mode of action.

## 4. Materials and Methods

### 4.1. Harvesting Xylem Sap Samples

To collect xylem sap exudate, the seeds of soybean cultivar C08 (*Glycine max*) were germinated in vermiculite under greenhouse conditions. One week after germination, the seedlings were transferred to a hydroponic culture in half-strength Hoagland solution, which was changed twice a week. Soybean plants were cultured to the V3 growth stage, in which the third trifoliates were fully unfolded. At the V3 growth stage, the xylem sap samples were collected via the root pressure method described by Djordjevic et al., with some modifications [49]. In brief, the plants were cut at the mid-section of the hypocotyl. The rootstock cutting wounds were then rinsed with MilliQ-grade water and gently wiped dry with tissue paper. Studies showed that the plants can rapidly seal off damaged sieve tubes through sieve element occlusion on phloem penetration or damage [50,51]. Therefore, to minimize the number of damaged cells and phloem sap contamination, the sap was allowed to bleed for the first 15 min to exude the damaged cells and ensure the natural healing of the phloem, and the bleeding wounds were then cleaned and wiped dry again. Pieces of plastic tubes that were 4 cm in length and with an inner diameter of 3 mm were then fitted on the rootstocks, with the gap between the stem and tubing sealed by wrapping a layer of hydrophobic parafilm around it. The xylem sap was allowed to exude via root pressure for a total of 6 h, and sap was periodically collected from the tubing using a syringe with a needle, and transferred to 1.5 mL tubes placed on ice for temporary storage. After collection, the sap samples were centrifuged and transferred to new tubes to remove any precipitates. Next, the sap samples were concentrated by vacuum-drying to <500 µL and were then frozen with liquid nitrogen and stored at −80 °C until downstream analysis.

### 4.2. Secreted Peptides’ Purification and Pre-Fractionation from Xylem Sap

Xylem sap peptides were purified and enriched using the *o-c*hlorophenol/acetone method with modifications [24]. Briefly, the xylem sap samples were equilibrated by N-ethylmorpholine (NEM) to total a concentration of 1%. *o-c*hlorophenol (1% NEM) was mixed with the equilibrated xylem sap by vortexing for 1 min. After centrifugation at 18,000× *g* for 5 min, the phenol phase was collected, and the extraction step was repeated twice. Immediately, the extracted phenol phase was supplemented with one-tenth the volume of 1% Ficoll as a co-precipitant, and was then precipitated with nine volumes of acetone overnight in a refrigerator set at −20 °C. The pellet was recovered by centrifugation at 18,000× *g* for 30 min at 4 ℃, followed by two ice-cold acetone washes with centrifugation at 18,000× *g* for 15 min. The obtained protein pellet was dried in vacuo and resuspended in water. Lastly, the sample solution was passed through an Amicon Ultra 10K MWCO filter (Merck Millipore, Carrigtwohill, Ireland) to collect the small peptide fraction as filtrate.

To discover xylem sap peptides by LC-MS/MS, the enriched peptide fraction was fractionated using a high-pH reversed-phase fractionation kit (Thermo Fisher Scientific, Rockford, USA). Briefly, the xylem sap peptide sample was first equilibrated to 0.1% trifluoroacetic acid (TFA) and then loaded on reversed-phase resin on spin columns. The bound peptides were sequentially eluted with a high-pH buffer (0.1% triethylamine) with increasing concentrations of acetonitrile (ACN): 7.5%, 12.5%, 17.5%, and 50%, divided into four fractions by centrifugation. The collected fractions were then vacuum-dried and resuspended in 0.1% formic acid (FA) before being loaded onto an LC column.

### 4.3. Peptidomic Analysis by Data-Dependent Nano-LC-MS/MS

The peptide samples were separated and analyzed with an Ultimate 3000 Nano-LC system (Dionex, Sunnyvale, USA) coupled with an Orbitrap Fusion Lumos Tribrid Mass Spectrometer (Thermo Fisher Scientific, Waltham, USA). Fractionated peptide samples were first quantified using the quantitative fluorometric peptide assay (Pierce, Thermo Fisher Scientific, Rockford, USA). Next, 0.5 µg of each peptide fraction was dissolved in 0.1% FA and injected into the Thermo C18 PepMap 100 trap reversed-phase column (300 µm × 5 mm, 5 µm particle size). Separation was achieved at 300 nL/min when using buffer A (0.1% FA) and buffer B (0.1%FA in ACN) as a mobile phase for gradient elution with a Thermo Acclaim PepMap RSLC C18 column (75 µm × 250 mm, 2 µm particle size) at 50 °C. Peptide elution employed a 90 min long-flow gradient method that started with an equilibration step of 0–5 min, followed by 0–30% buffer B for 5–70 min, a washing step of 80% buffer B for 70–80 min, 80–0% buffer B for 80–80.1 min, and 0% buffer B for 80.1–90 min. The duty cycle was set for MS1 scanning at a 60,000 resolution in the range of 375–1500 m/z followed by a 3-s-long data-dependent MS2 scan at a 15,000 resolution using higher-energy collisional dissociation (HCD) fragmentation. Dynamic exclusion of precursor ions was set to 60 s with mass tolerance of 10 ppm.

The generated MS raw data were analyzed using Proteome Discoverer (version 2.4). The MS files were searched against a soybean database with the enzyme set as “no-enzyme”. Methionine oxidation, proline oxidation, tyrosine sulfation, and proline triarabinosylation were set as variable modifications. Precursor and fragment ion mass tolerances were set to 10 ppm and 0.02 Da, respectively. The minimum peptide length was set to 5, and the maximum length was set to 30. Peptide validation was performed under “percolator” based on q-values with FDR <0.05.

### 4.4. Quantitative Reverse Transcription PCR

For tissue expression pattern analysis, the soybean seeds were germinated in wet vermiculite under greenhouse conditions. One week after germination, the seedlings were transferred to a hydroponic culture in half-strength Hoagland solution. When the soybean plants reached the V1 growth stage, in which the first trifoliates were fully unfolded, total RNA was collected from the leaves, second internode (between the first trifoliate and unifoliate), hypocotyl stem, and root tissues using an RNeasy Plant Mini Kit (Qiagen, Hilden, Germany), and cDNA was synthesized using the QuantiTect Reverse Transcription Kit (Qiagen, Hilden, Germany) following the manufacturer’s protocols. The primers were generated with NCBI online Primer-BLAST against the *Glycine max* genome. qPCR was performed using SYBR^TM^ Select Master Mix (Applied Biosystems, Vilnius, Lithuania) and conducted on a CFX96 System (Bio-Rad, Hercules, CA, USA). The PCR condition was set as follows: a UDG activation step at 50 ℃ for 2 min and then a denaturing step at 95 ℃ for 2 min, followed by 45 cycles of 95 ℃ for 15 s, 57 ℃ for 15 s, and 72 ℃ for 1 min. For the tissue expression study, the soybean *ELF1b* gene was used as a reference gene for normalization. For abiotic stress response analysis, the soybean *Bic-C2* gene was used as a reference gene for normalization [52].

### 4.5. Stress Treatment

For stress-response analysis, soybean cultivar C08 seeds were germinated in vermiculite under greenhouse conditions. One week after germination, the seedlings were transferred to a hydroponic culture in a nutrient solution comprising 2.5 mM KNO_3_, 1 mM MgSO_4_, 0.5 mM KH_2_PO_4_, 2.5 mM CaCl_2_, 30 µM Fe-EDTA, 46.25 µM H_3_BO_4_, 9.1 µM MnCl_2_, 0.765 µM ZnSO_4_, 0.32 µM CuSO_4_, and 0.5 µM H_2_MoO_4_. At the V1 growth stage, in which the first trifoliates were fully unfolded, all of the plants were transferred to control or treatment nutrient solutions for a 48-h-long treatment. The control plants were cultured in nutrient solution without modification. For the nitrogen-, phosphate-, and iron-deficiency conditions, the treated plants were cultured in a nutrient solution without KNO_3_, KH_2_PO_4_, or Fe-EDTA, respectively. For the high-nitrogen supply treatment, the KNO_3_ concentration was increased by 10-fold, to 25 mM, in the nutrient solution for the treated plants. For the salinity treatment, the nutrient solutions for the treated plants were supplemented with 0.6% NaCl. For the PEG-induced drought treatment, the nutrient solution was supplemented with 10% PEG 6000 to induce water-deficit stress. After 48 h of treatment, the root tissues of the plants were harvested, frozen with liquid nitrogen, and stored at −80 °C until RNA extraction for cDNA synthesis.

For the PRM assay, 7 days after germination, the soybean plants were transferred to a hydroponic culture supplied with the nutrient solution described above, and the solution was changed twice a week. At the V3 growth stage, when the third trifoliates were fully unfolded, the soybean plants were then treated with either normal nutrient solution or a nitrogen-depleted solution for 48 h. Xylem sap was then collected from the plants by the root pressure method described above, and the collected sap samples were frozen with liquid nitrogen and stored at −80 °C until downstream analysis.

### 4.6. Biological Activity Assay of Modified Peptides

The synthetic peptides with hydroxylated proline used in the experiment were ordered from GL Biochem Ltd. (Shanghai, China) and had a purity >95%. Four days after germination, soybean seedlings were transferred to half-strength Hoagland’s solution either with or without synthetic peptides (1 µM) for a 7-day-long treatment under greenhouse conditions with natural light. Treated plant roots were measured to determine the root growth conditions.

### 4.7. Peptide Quantification by LC-PRM Assay

MS analysis was performed with an Ultimate 3000 nano-LC system connected to a Orbitrap Fusion Lumos Tribrid mass spectrometer, as stated before. Peptide elution followed a 60 min-long flow gradient with an equilibration step for 0–8 min, 0–20% buffer B for 8–50 min, 20–80% buffer B for 50–51 min, 80% buffer B for 51–55 min, 80–0% for 55–55.5 min, and 0% buffer B for 55.5–60 min. Synthetic peptides of GmCEP-XSP1/GmXAP6a-D1, GmCEP-XSP1-D2, and GmXAP6a-D2 carrying different degrees of proline hydroxylation were first analyzed under the DDA mode to obtain the MS/MS spectra. The raw data were searched against the database using Proteome Discoverer. The peptide-matching result files were then imported into Skyline software (version 21) for ion chromatogram extraction and used to generate a peptide library [30]. The peptide precursor/fragment ion (transitions) pairs were manually examined in a Skyline document, and the most dominant precursor charge state and three most intense and consistent fragment ions were selected for each peptide to construct a precursor isolation list (Appendix A).

Small peptide samples were extracted from the xylem sap collected from the low-nitrogen treatment experiment using the *o-c*hlorophenol/acetone method, as previously mentioned. Purified and enriched xylem sap peptides were first quantified using the quantitative fluorometric peptide assay (Pierce, Thermo Fisher Scientific, Rockford, USA). Then, equal amounts of the peptide samples (0.75 µg) from the untreated and nitrogen-starved soybean xylem sap were injected into a column in each run. Xylem sap peptide samples were analyzed by a mass spectrometer under the target MS mode following the assigned precursor isolation list. The precursor ions of the targets were monitored in an unscheduled PRM setting. The duty cycle was first set as an MS1 scan at 120,000 resolutions in the 400–1500 m/z range and with a maximum injection time of 246 ms. The MS1 scan was followed by a targeted MS2 scan targeting a pre-assigned precursor isolation list. The MS2 scan was performed with the following parameters: isolation window of 1.4 m/z; Orbitrap resolution of 60,000; mass range set by “define first mass” to 100 Da; maximum injection time of 118 ms; loop control setting of “All”. The raw MS files that were obtained were first searched against the database in Proteome Discoverer for peak identification. The peptide-matching result files (.msf) and raw MS data were then processed by the Skyline software to generate extracted ion chromatograms and to perform peak integration using the synthetic peptide library. The MS2 peak areas were generated by the three most intense fragment ions (transitions), and the coefficient of variation (CV) values for each individual fragment ion were calculated within the technical replicates (*n* = 3) of each sample.

### 4.8. Hairy Root Transformation

Target genes were first cloned and inserted into entry vector pENTR/D-TOPO (Invitrogen, Thermo Fisher Scientific, Carlsbad, USA) according to manufacturer’s protocols, and then transformed into destination vector pGWB2 using the Gateway LR Clonase Enzyme mix (Invitrogen, Thermo Fisher Scientific, Carlsbad, USA). The resulting pGWB2 gene constructs were transformed into *Agrobacterium rhizogenes* K599, and empty vector pGWB2 was used as a control. For hairy root induction, 4 days after soybean seed germination, the seedlings were infected with a transformed K599 culture via needle injection into the hypocotyl region. The original roots of the infected plants were removed 2 weeks after infection, leaving only the induced hairy roots. The plants were then transferred to a hydroponic culture with half-strength Hoagland’s solution for another 2 weeks for hairy root development. Afterward, root tissues were harvested and frozen with liquid nitrogen before protein extraction.

### 4.9. Total Protein Extraction and Tryptic Digestion

Hairy root tissues from soybean transformants were collected 4 weeks post-infection and ground into fine powder in liquid nitrogen. Total protein extraction was performed via the SDS/phenol extraction method [53]. In brief, 100 mg of root tissues was incubated in 0.9 mL of TCA/acetone buffer (1:8) for 30 min at −20 °C. Tissue pellets were precipitated by centrifugation at 18,000× *g* for 15 min at 4 °C. The supernatant was discarded, and the tissue pellets were washed twice with chilled acetone, with each wash followed by centrifugation at 18,000× *g* for 15 min at 4 °C. The tissue pellets were briefly dried in vacuo. Then, the dried pellets were incubated in SDS extraction buffer (1% SDS, 0.15 M Tris-HCl (pH 8.8), 1 mM EDTA, 100 mM DTT, and protease inhibitors (cOmplete ULTRA Tablets, EDTA-free, Roche, Mannheim, Germany) for 15 min at room temperature, and the mixture was centrifuged at 18,000× *g* for 10 min at 4 °C to collect the supernatant in new tubes. The extraction step was repeated once, and the SDS buffer extracts were combined. The SDS extract was mixed with an equal volume of phenol (pH 8.0) by vortexing for 5 min. The phenol phase was collected by centrifugation at 18,000× *g* for 10 min at 4 °C. The collected phenol phase was then cleaned up with wash buffer (100 mM Tris-HCl, pH 8.0, 0.7 M sucrose, 1 mM EDTA) by vortexing for 5 min, followed by 10 min of centrifugation to collect the phenol phase. Finally, the phenol phase containing the proteins was precipitated by nine volumes of 0.1 M ammonium acetate in methanol for 30 min at −20 °C, then was subsequently centrifuged at 18,000× *g* for 15 min at 4 °C to obtain pellets. The pellets were washed again with 0.1 M ammonium acetate in methanol, and then twice with 80% acetone. Lastly, the protein pellet was air-dried andresuspended in 8 M urea and 10 mM ammonium bicarbonate (AB) for the downstream workflow.

Tryptic digestion of proteins was performed by the filter-aided sample preparation (FASP) method [54]. In brief, 50 µg of protein samples dissolved in 8 M urea and 10 mM AB buffer ware filtered through an Amicon Ultra 10K MWCO filter (Merck Millipore, Carrigtwohill, Ireland) by centrifugation at 12,000× *g* for 10 min to remove salt and small molecules. The proteins retained on the filter were then reduced and alkylated with DTT and IAA in urea buffer, respectively, and the reagents were removed by centrifugation. Buffer exchange was performed with 10 mM AB by carrying out successive centrifugation three times. Finally, the protein samples retained on the filter were digested overnight with 1 µg sequencing-grade modified trypsin (Promega, Madison, USA) at 37 °C, and the yielded peptide solution was collected from the filter by reverse spinning, before being dried in vacuo. Finally, the peptide samples were resuspended in 300 µL of 0.1% TFA and were further purified by using Peptide Desalting Spin Columns (Pierce, Thermo Fisher Scientific, Rockford, USA). The elutes were dried and then resuspended in 0.1% FA for LC-MS.

### 4.10. Label-Free Quantitative Proteomic Analysis

The LC-MS system for label-free proteomic analysis was set up under the same conditions as stated above. Tryptic peptides were first quantified using the quantitative fluorometric peptide assay (Pierce, Thermo Fisher Scientific, Rockford, USA). Peptide samples of 0.75 μg resuspended in 0.1% FA were injected for each technical run. Peptide elution employed a 180 min-long gradient comprising an equilibration step for 0–5 min, 0–35% buffer B for 5–160 min, 35–80% buffer B for 160–165.5 min, a washing step of 80% buffer B for 165.5–175 min, 80–0% for 175–176 min, and 80–0% buffer B for 175–180 min. The duty cycle was set with an MS1 scan at a 120,000 resolution in the range of 300–1800 m/z, followed by a 3-s data-dependent MS2 scan at 30,000 resolution using HCD for fragmentation. The dynamic exclusion of precursor ions was set to 30 s with 10 ppm mass tolerance.

Label-free quantitative proteomic data analysis was performed with Proteome Discoverer, and the MS/MS spectra were searched against the soybean database with the enzyme set as trypsin, with one missed cleavage allowed. Carbamidomethyl cysteine was set as a fixed modification, and oxidized methionine and proline were set as variable modifications. Precursor and fragment ion mass tolerances were set to 10 ppm and 0.02 Da, respectively. The minimum peptide length was set to 6, and the maximum length was set to 30. Peptide validation was performed under “percolator” based on q-values with FDR <0.05. Identified proteins were filtered for quantification as follows: unique peptides greater than or equal to 1; “Confidence” achieved at least “Peak Found” in all samples and “High” in at least two samples. The statistical significance of the differentially expressed proteins was analyzed using the Benjamini–Hochberg method, where the protein abundance ratio between control and treatment groups with adjusted *p*-values <0.05 was considered to determine DEPs.

### 4.11. GO Term Analysis

The GO annotations of the proteins from the label-free quantification analysis were obtained through the SoyBase (https://www.soybase.org, accessed on 11 March 2022) tool for GO term enrichment [55]. We performed our GO enrichment analysis via the Fisher’s Exact Test function in Blast2GO Basic by comparing the GO annotations of the DEPs and the total number of identified proteins [56]. The enriched GO terms were filtered by FDR <0.05. The image of the hierarchy tree of the enriched GO terms was generated by Blast2GO Basic based on the GO enrichment results with the filtering conditions FDR <0.01 and matching at least 20 proteins.

## Figures and Tables

**Figure 1 ijms-23-08641-f001:**
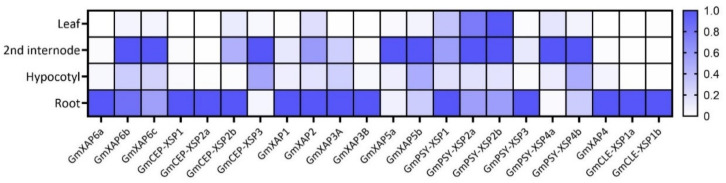
Relative expression levels of soybean xylem sap peptide precursor genes in different tissue regions. The relative expression levels of the peptide precursor genes are based on the tissues with maximal expression. Values of 1 represent the maximal expression level of the gene. The soybean gene *ELF1b* was used as a reference gene for normalization.

**Figure 2 ijms-23-08641-f002:**
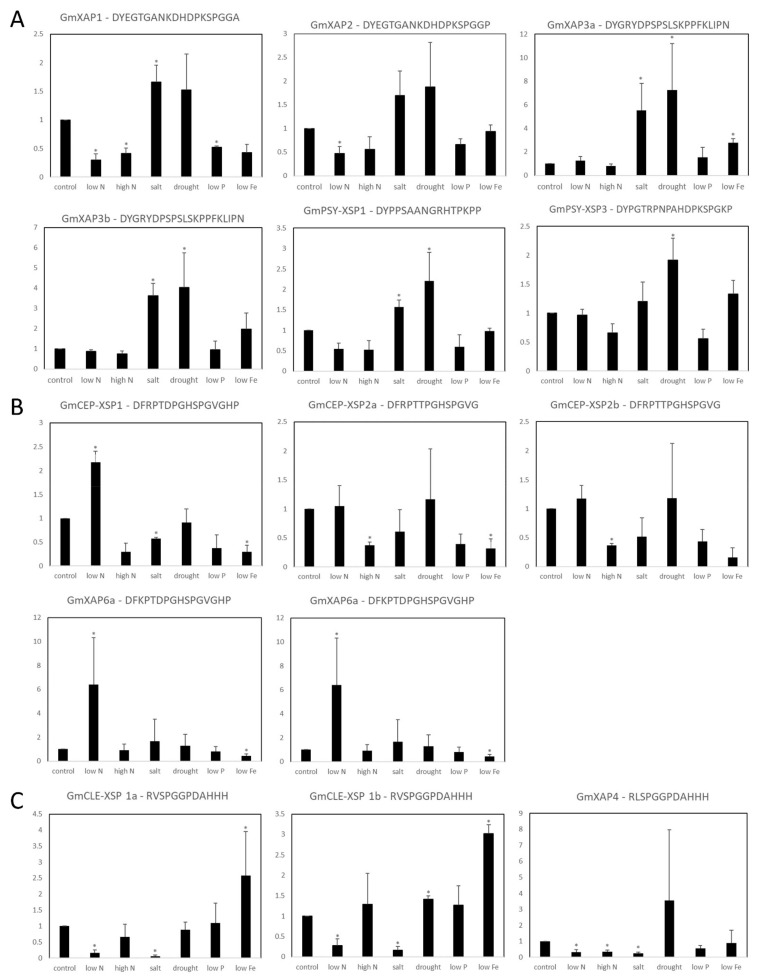
Relative expression levels of xylem sap peptide precursor genes in soybean roots under various abiotic stress conditions: (**A**) tyrosine-sulfated peptides; (**B**) C-terminally encoded peptides (CEPs); (**C**) CLAVATA3/embryo surrounding region-related peptides (CLEs). The soybean gene *Bic-C2* was used as a reference gene for normalization. The relative expression values are based on expression in the control plants. The error bars indicate standard deviation (*n* = 3). Statistical significance was evaluated by Student’s t-test, and significant differences (*p*-value < 0.05) between the treatment and control plants are indicated by asterisks (*).

**Figure 3 ijms-23-08641-f003:**
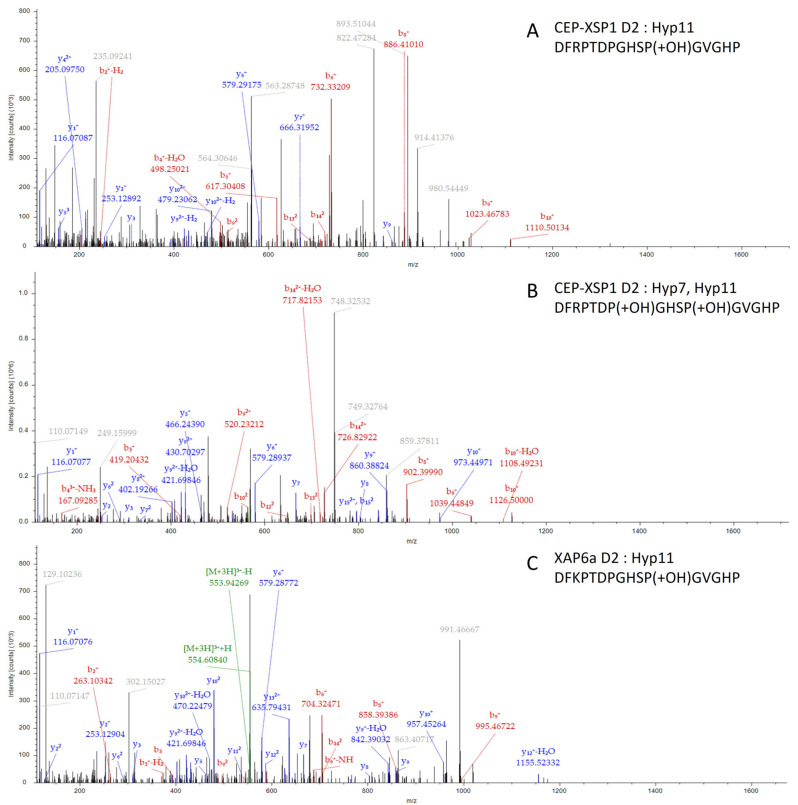
MS/MS spectra of CEP-XSP1 D2 (Hyp11), CEP-XSP1 D2 (Hyp7, Hyp11), and XAP6a (Hyp11) identified in peptide extract from soybean xylem sap: (**A**) MS/MS spectrum of CEP-XSP1 D2 peptide with hydroxylation at P11 (m/z 563.59784, z = 3); (**B**) MS/MS spectrum of CEP-XSP1 D2 peptide with hydroxylation at P7 and P11 (m/z 568.92926, z = 3); (**C**) MS/MS spectrum of XAP6a D2 peptide with hydroxylation at P11 (m/z 554.26093, z = 3).

**Figure 4 ijms-23-08641-f004:**
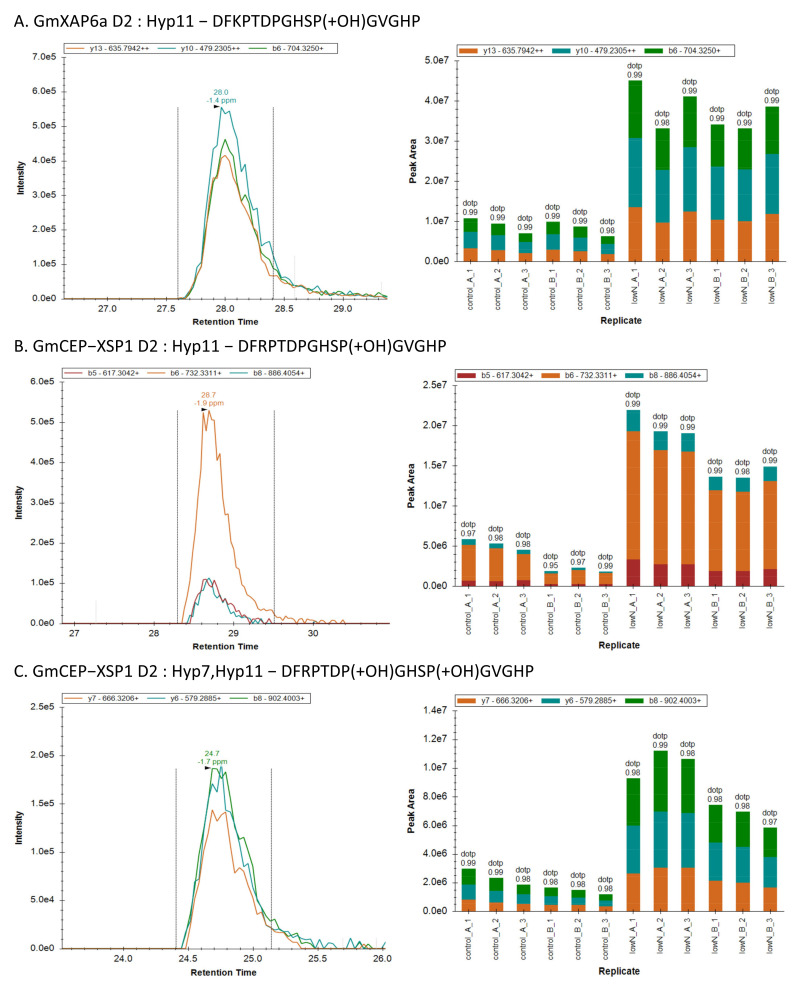
Parallel reaction monitoring (PRM) assay of peptides: (**A**) GmXAP6a:Hyp11, (**B**) GmCEPXSP1:Hyp11, and (**C**) GmCEPXSP1:Hyp7,Hyp11 in soybean xylem sap under normal (control) and nitrogen-deficiency (Low N) conditions. The left panels show the extracted ion chromatogram of the three transitions (fragment ions) selected for quantification. The right panels show the peak areas of each technical replicate plotted in the bar chart. The experiment was performed in biological replicates (**A**,**B**) and technical triplicates (1–3) for each condition.

**Figure 5 ijms-23-08641-f005:**
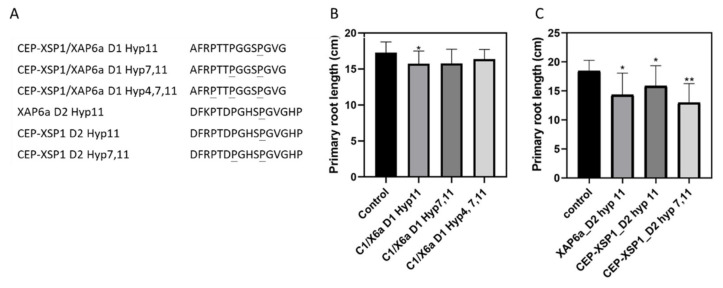
Biological activity of synthetic peptides of CEP-XSP1 and XAP6a: (**A**) modified peptide sequences of CEP-XSP1/XAP6a domain 1 (D1), XAP6a domain 2 (D2), and CEP-XSP1 domain 2 (Hyp = proline hydroxylation, underlined = hydroxylated sites); (**B**) primary root lengths of soybean seedlings after 7 days of treatment with CEP-XSP1/XAP6a-D1 peptides (shown as C1/X6a); (**C**) primary root lengths of soybean seedlings after 7 days treatment with XAP6a-D2 and CEP-XSP1-D2 peptides. To assess the peptides’ biological activity, 4-day-old soybean seedlings (*n* = 20) were cultured with or without 1 µM peptides in nutrient solution for 7 days, and the primary root length of treated plants was then measured. A statistical difference was assessed by Student’s *t*-test (* = *p*-value < 0.05, ** = *p*-value < 0.01). Error bars indicate the standard deviation.

**Figure 6 ijms-23-08641-f006:**
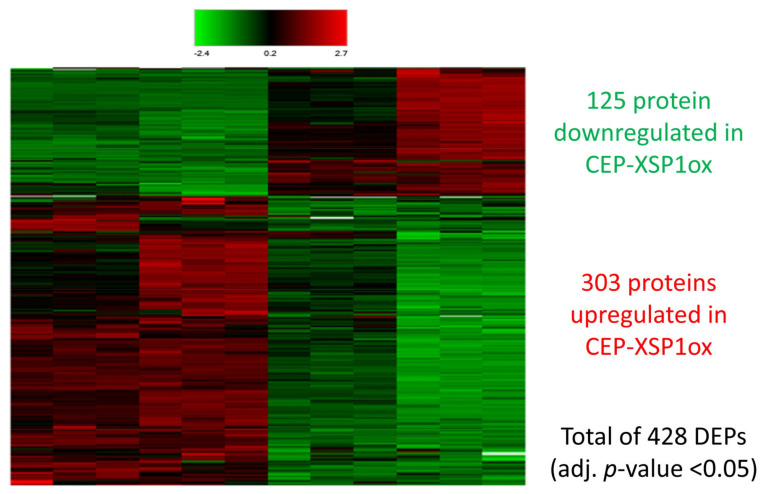
Heatmap representing clustering of differentially expressed proteins (DEPs) in control and CEP-XSP1ox hairy root identified in label-free proteomic quantification analysis following LC-MS/MS. Statistical significance of quantified proteins was analyzed by the Benjamini–Hochberg method, with proteins with adjusted *p*-values < 0.05 considered as DEPs. Relative fold changes are indicated on the log2 scale. Numbers of down- and upregulated proteins in CEP-XSP1ox are indicated in green and red, respectively.

**Figure 7 ijms-23-08641-f007:**
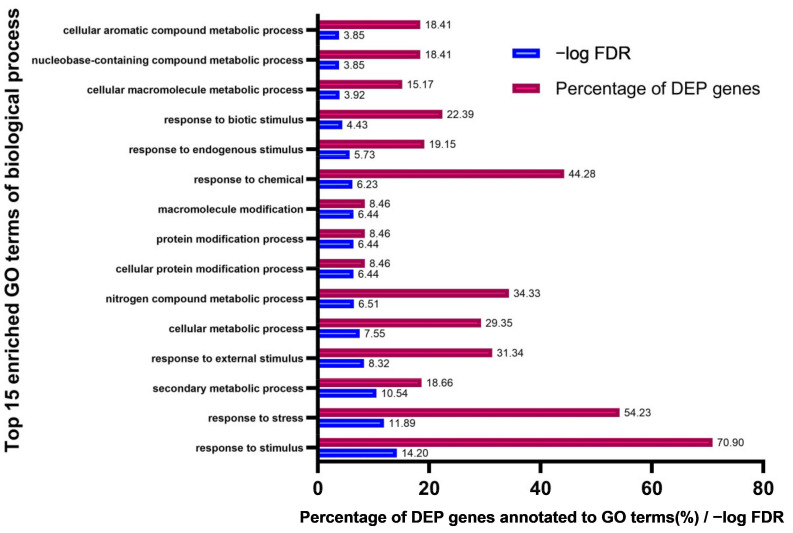
Top15 enriched gene ontology (GO) terms of biological processes from the 428 differentially expressed proteins (DEPs) in the CEP-XSP1ox soybean hairy root proteome. The enriched GO terms were ranked by their -log FDR value, which is indicated with the blue bar. The percentage of genes annotated to the GO terms relative to the total number of genes in the input list is indicated with the red bar (%).

**Figure 8 ijms-23-08641-f008:**
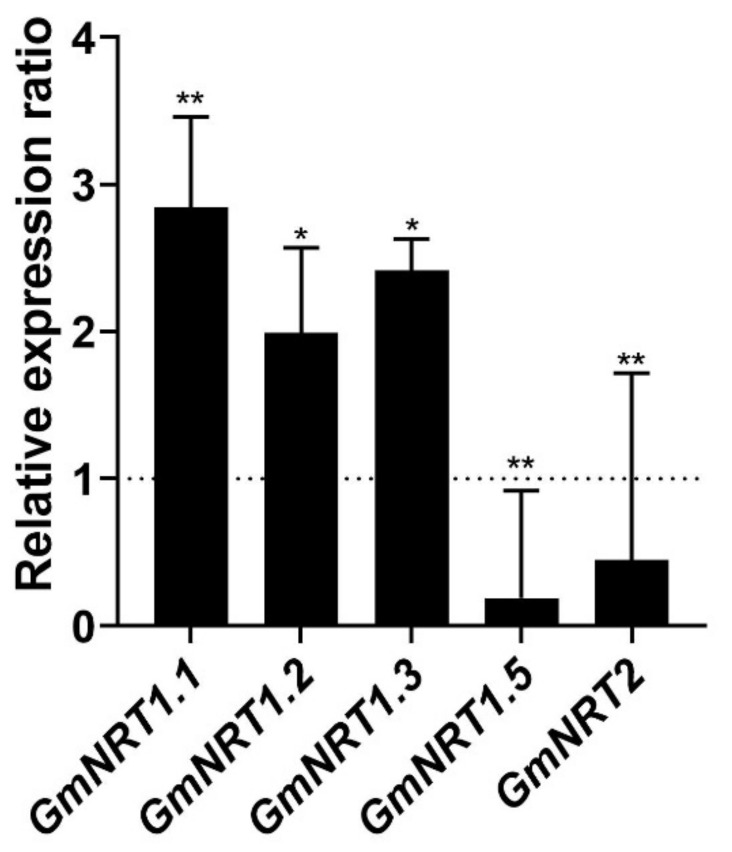
Relative expression ratio of *NRT* genes from the *GmCEP-XSP1ox* line to the control line in soybean hairy root tissues. The error bars indicate standard deviation (*n* = 4). The expression level is relative to the control hairy root (empty vector). Significant difference was analyzed by Student’s *t*-test (*= *p*-value < 0.05, ** = *p*-value < 0.01).

**Table 1 ijms-23-08641-t001:** Identification of plant-secreted peptide sequences from soybean xylem sap by LC-MS/MS. The peptide sequences are categorized based on peptide family classification into CEP, PSY and CLE: D1 = domain 1; D2 = domain 2; Hyp = hydroxylated proline; S = sulfation; Tar = tri-arabinosylation; Nm = no modification. ^1^ Overlapping sequences between GmXAP6a and GmCEP-XSP1. ^2^ Overlapping sequences among GmXAP6a–c. * Full length of proposed peptide domain not detected in the MS spectra.

Gene ID	Peptide Groups	Observed Peptide Domain Length Variants	Peptide PTM Positions and Combinations	Proposed Final Domains
Glyma.01G185000.1	GmXAP6a	D1: AFRPTTPGNSPGVG ^1^	Hyp7,11; Hyp4,7,11	D1: AFRPTTPGNSPGVGHK
		D1: AFRPTTPGNSPGVGHK ^1^	Hyp4,7,11	D2: DFKPTDPGHSPGVGHP
		D2: DFKPTDPGHSPGVG ^2^	Hyp11	
		D2: DFKPTDPGHSPGVGH ^2^	Hyp11	
		D2: DFKPTDPGHSPGVGHP	Hyp11	
Glyma.05G084100.1	GmXAP6b	D2: DFKPTDPGHSPGVG ^2^	Hyp11	* D1: AFRSTTPGGSTGVGHK
		D2: DFKPTDPGHSPGVGH ^2^	Hyp11	D2: DFKPTDPGHSPGVGHV
		D2: DFKPTDPGHSPGVGHV	Hyp11	
Glyma.17G177300.1	GmXAP6c	D1: AFRPTTPGGSPGVG	Hyp4,7,11	* D1: AFRPTTPGGSPGVGHK
		D2: DFKPTDPGHSPGVG ^2^	Hyp11	* D2: DFKPTDPGHSPGVGHA
		D2: DFKPTDPGHSPGVGH ^2^	Hyp11	
Glyma.11G057200.1	GmCEP-XSP1	D1: AFRPTTPGNSPGVG ^1^	Hyp7,11; Hyp4,7,11	D1: AFRPTTPGNSPGVGHK
		D1: AFRPTTPGNSPGVGHK ^1^	Hyp4,7,11	D2: DFRPTDPGHSPGVGHP
		D2: DFRPTDPGHSPGVGHP	Hyp11; Hyp7,11	
Glyma.05G083900.1	GmCEP-XSP2a	DFRPTTPGHSPGVG	Hyp4,7,11	DFRPTTPGHSPGVG
Glyma.17G176500.1	GmCEP-XSP2b			
Glyma.06G037700.1	GmCEP-XSP3	DFQPTDPGHSPGAGH	Hyp11	DFQPTDPGHSPGAGHSSPH
		DFQPTDPGHSPGAGHSSPH	Hyp4,7,11	
Glyma.19G117800.1	GmXAP1	DYEGTGANKDHDPKSPGGA	Nm; Hyp16; Y2S, Hyp16	DYEGTGANKDHDPKSPGGA
Glyma.16G035200.1	GmXAP2	DYEGTGANKDHDPKSPGGP	Y2S; Hyp16; Y2S, Hyp16	DYEGTGANKDHDPKSPGGP
Glyma.13G201100.1	GmXAP3a	DYGRYDPSPSLSKPP	Hyp7,9; Hyp9; Hyp9,14	DYGRYDPSPSLSKPPFKLIPN
Glyma.12G235600.1	GmXAP3b	DYGRYDPSPSLSKPPF	Hyp7,9; Hyp9; Hyp9,14	
		DYGRYDPSPSLSKPPFK	Hyp9; Hyp9,14; Y2S, Hyp9,14; Y2S, Hyp7,9,14	
		DYGRYDPSPSLSKPPFKLIPN	Hyp9,14	
Glyma.04G233800.1	GmXAP5a	DYDEAGPNPRHT	Nm, Y2S	DYDEAGPNPRHTKKPGKG
Glyma.06G131000.1	GmXAP5b	DYDEAGPNPRHTKKPGKG	Y2S	
Glyma.06G194600.1	GmPSY-XSP1	DYPPSAANGRHTPKPP	Hyp4,13; Y2S, Hyp4,13; Y2S, Hyp4,13, P3Tar; Y2S, Hyp13, P4Tar	DYPPSAANGRHTPKPPY
		DYPPSAANGRHTPKPPY	Y2S, Hyp4,13,15	
Glyma.03G007400.1	GmPSY-XSP2a	DYPGTGANNRHDPKTPGGP	Y2S, Hyp16	DYPGTGANNRHDPKTPGGP
Glyma.19G118700.1	GmPSY-XSP2b			
Glyma.19G117900.1	GmPSY-XSP3	DYPGTRPNPAHDPKSPG	Y2S, Hyp16	DYPGTRPNPAHDPKSPGKP
		DYPGTRPNPAHDPKSPGKP	Y2S, Hyp16	
Glyma.05G219100.1	GmPSY-XSP4a	DYKESGANPRNDQGKTKPHG	Nm; Y2S	DYKESGANPRNDQGKTKPHG
Glyma.08G025400.5	GmPSY-XSP4b			
Glyma.13G171400.1	GmXAP4	RLSPGGPDAHHH	Hyp4,7; Hyp4, P7Tar	RLSPGGPDAHHH
Glyma.07G204600.1	GmCLE-XSP1a	RVSPGGPDAHHH	Hyp4,7; P4Tar, Hyp7	RVSPGGPDAHHH
Glyma.13G171500.1	GmCLE-XSP1b			

## Data Availability

The data presented in this study are available on request from the corresponding author.

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
