# Peer review of "Identification of Diverse Stress-Responsive Xylem Sap Peptides in Soybean"

_ijms, 2022, doi:10.3390/ijms23158641_

Round 1

Reviewer 1 Report

In this manuscript, the authors purified small peptides from soybean xylem sap and identified their post-translational modification. Furthermore, some of these peptides were demonstrated to be related to stress responses. The authors proposed that these post-translational modified oleo-peptides might be involved in the biological functions and mechanisms of long-distance signaling in plants. The data are scientifically sound and the proposed physiological meaning is convincing. Some comments are listed below for the consideration in revision.

1.      The resolution of Figures 2 and 4 is poor. Please improve it.

 2.      Please check for the significance in Figure 2. For example, GmXAP3b showed up-regulation in drought condition but not in salt? The expression level of GmXAP3b seemed to be varied from the control in salt and low Fe condition as well. Please provide some explanations.

 3.      The xylem gap samples were collected from the four-week-old soybean but the total RNA was extracted from a 20-day old plant. The abiotic stresses were even conducted on a 15-day old soybean plant. For PRM assay, a 28-day old plant was treated. The plant conditions in different experiments were different, and different stages of plants might greatly affect the signal transduction in plants. Please explain why different stages of plants were chosen in their detections.

 4.      The soybean plants were cultured in a hydroponic condition in this study. The root morphology, phenotypes and signaling in plants differ greatly from the plants grown in soil condition. Will the results observed in this study be applicable to plants grown in field?

Reviewer 2 Report

The manuscript describes the study that deals with plant secretory peptides, a part of long-distance signaling in plants. Authors developed methods to enrich and purify small peptides from soybean xylem sap and determined primary structures of 14 peptide groups belonging to different families.  In general, it was demonstrated that the peptides take part in regulating stress adaptation processes.

The work performed is systematic and thorough, and the conclusions reached by the authors are warranted by the data obtained. The manuscript is well-written and can be published in the present form.

Author Response

Dear reviewer,

Thank you for reviewing our manuscript, your positive comments on our manuscript provided encouragement and support for our team members.

Your sincerely,

Mr. SIN Wai Ching, Prof. LAM Hon Ming and Prof. NGAI Sai Ming